# PROMPT TUNING IS ALL WE NEED?

## ABSTRACT

Recent advances in pre-trained vision-language models, e.g., CLIP, have demonstrated remarkable success in domain generalization (DG) by tuning prompts. To promote DG, one promising method is to explore how to design or learn more sweet prompts, i.e., prompt learning. The implicit intuition of it is that a more elaborate prompt learning method can lead to higher generalization performance. The foundation intuition motivates us to raise a question: *Prompt tuning is all we need?* To verify whether the intuition holds for DG, we design comprehensive experiments on DG benchmarks. However, our experiments demonstrate a pessimistic conclusion that simply tuning prompts using training sets can achieve comparable performance with that using test sets. Namely, even the optimal prompts can hardly bring significant performance gain than a simple tuning strategy. Our experiments show that this results from the non-separability of features extracted by the image encoder. Thus, we propose *im*age encoder tuning, named Im-Tuning, for more separable image features. We conduct extensive experiments on multiple DG benchmarks, demonstrating that Im-Tuning can consistently outperform the relevant state-of-the-art methods.

## 1 INTRODUCTION

Contrastive Language-Image Pretraining (CLIP) has shown remarkable capabilities in many downstream tasks (Radford et al., 2021), particularly in the field of domain generalization (DG), which is a challenging task that aims to enable models to generalize well across different domains. Existing CLIP-based DG methods (Zhou et al., 2021; 2022a) mainly focus on learning (or designing) proper text prompts to extract the potential generalization ability of the CLIP model. The implicit intuition of the prompt learning strategy is that more proper prompts may result in better alignments between image and text features, which is consistent with the pre-training process of CLIP models.

The intuition motivates us to raise a question: is prompt tuning all we need for DG? Namely, we challenge the necessity of optimal prompts learned for DG. In particular, we conduct comprehensive experiments using DG benchmarks, e.g., DomainNet (Peng et al., 2019) and Office-Home (Venkateswara et al., 2017), to investigate the priority of optimal prompts than a hand-crafted prompt. Our experimental results demonstrate that *simply tuning prompts using training sets can achieve comparable performance with that using test sets.* That is, slightly tuning prompts can achieve comparable performance with optimal prompts, which significantly weakens the motivation for optimal prompts and guides us to explore new fine-tuning methods.

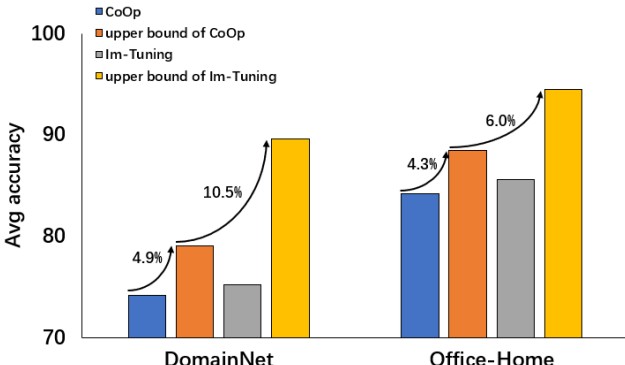

Figure 1: We conduct a small experiment to validate if a simple prompt turning strategy (e.g., CoOp) is suitable in multi-source DG datasets. Figure 1(a) illustrates the results on DomainNet. Figure 1(b) illustrates the results on Office-Home.

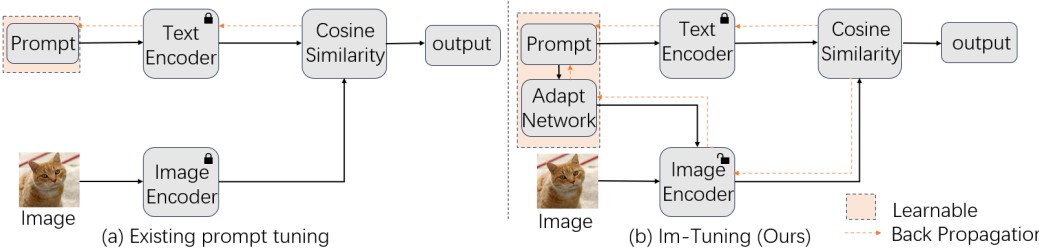

Figure 2: The concept illustration of two different methods to apply CLIP in DG. (a) The prompt tuning method only trains the prompt and freezes the visual backbone of CLIP. (b) Our Im-Tuning method enables gradients to propagate jointly in both modalities to enhance their synergy.

Fortunately, our experiments have demonstrated that the optimal prompts do not achieve the desired performance due to the inherent inseparability of image features. Taking DomainNet as an example, the utilization of data exclusively from a single target domain, along with a simple prompt turning strategy (e.g., CoOp (Zhou et al., 2022b)), results in the derivation of an optimal prompt that only achieves an average accuracy of 79.3%. This improvement is only 4.9% higher than the accuracy achieved by training the model using data from the source domain. This limitation can be attributed to the indivisibility of image features, which hampers the ability to achieve higher accuracy. Subsequently, this insight has inspired us to shift our focus to the image encoder and explore methods to enhance the separability of image features.

To make image features more separable, we propose tuning the image encoder, named Im-Tuning. Fine-tuning the entire image encoder is one straightforward approach. However, due to the limited availability of data, fine-tuning the entire model can easily disrupt its intrinsic information, leading to performance degradation. Consequently, we have been inspired to explore the adjustment of certain crucial statistical properties within the image encoder, i.e., scale and bias. Additionally, fine-tuning the image encoder alone may introduce difficulties in aligning image and text features across the entire model. To tackle this challenge, we propose utilizing prompts to generate scale and bias, which influence the image features. Im-tuning is a novel method that enables gradient propagation through both image and text encoders, thereby reducing the gap between fine-tuned image encoders and frozen text encoders, which is illustrated in Figure 2.

Through experimental validation, Im-tuning has been shown to acquire more separable image features, leading to improved generalization performance. This is particularly evident with the optimal prompt training on the test set, i.e., upper bound performance. Specifically, our method exhibits an average improvement of 8.25% compared to CoOp on the upper bound performance. Subsequently, we conducted experiments on three benchmark domain generalization datasets, where our method demonstrates a performance improvement of over 1% when compared to alternative methods.

Our main contributions are summarized as follows:

- Our work challenges the necessity of prompt tuning, and shows that an optimal prompt provides limited performance gain compared to a simple prompt. This can be attributed to the separability of image features, which suggests a promising direction for promoting domain generalization with clip models.
- We introduce a new method called Im-Tuning that manipulates the image features by modifying the parameters of the image encoder. To bridge the image-text modalities, Im-Tuning updates the parameters of the image encoder by the learned text prompt, enabling the bidirectional propagation of gradients and enhancing synergy.
- Comprehensive experiments conducted on various datasets demonstrate that Im-Tuning outperforms baseline methods, achieving state-of-the-art performance.

## 2 RELATED WORK

**Domain generalization.** The DG problem has various types. For example, multi-source DG (Gulrajani & Lopez-Paz, 2020) assumes multiple training domains, while single-source DG (Peng et al.,

2022) assumes one. Most DG techniques tackle the domain shift issue in a closed-set setting , where all domains have the same label set. Some methods address the more difficult heterogeneous DG (Wang et al., 2020), which has different labels between source and target domains.

More recently, Zhang et al. (2021b) evaluate the performance of CLIP (Radford et al., 2021) on data from multiple domains and proposed a domain prompt generator, where domain information is inferred by averaging the batch-wise visual features from CLIP's vision encoder. Besides using the domain information, the method in Zhang et al. (2021b) has some drawbacks: i) it captures domain information from the final layer of the vision encoder, which encodes more semantic information than domain-specific artifacts; ii) it works well with large batch sizes and may overfit for small batches since estimating style from a small number of feature vectors is not feasible. Recent works (Niu et al., 2022) learn domain-invariant prompts by using text-based source domain knowledge or image patches to learn the prompt input for ViT models, similar to visual prompt tuning (VPT) (Jia et al., 2022). Another recent line of research (Mancini et al., 2020) define the problem of visual recognition for unknown domains and classes, combining DG with zero-shot learning (Xian et al., 2017). Subsequent works use multimodal information (Chandhok et al., 2021) or disentangle feature learning (Mangla et al., 2021) for this purpose.

**Prompt learning.** Foundation models learn semantically rich visual representations by using text information and become prevalent in language processing (Devlin et al., 2018) and computer vision (Bommasani et al., 2021). The success of these models depends on the design of task-centric textual descriptions for visual data (Henaff, 2020). While earlier prompting strategies were mostly manual, subsequent works use prompt learning. For example, CoOp (Zhou et al., 2022b) optimize unified and class-specific prompts in the continuous space with back propagation. To improve CoOp's generalization, CoCoOp (Zhou et al., 2022a) use input-conditioned prompt learning. While CoOp Zhou et al. (2022b) learn projectors for textual input, CLIP-Adapter (Gao et al., 2021) fine-tunes feature adapters in both visual and language branches. ProGrad (Zhu et al., 2022) use a similar method to CoCoOp, preserving the knowledge learned from the foundation model. In Shu et al. (2022), test time prompt tuning (TPT) is proposed, where consistency among multiple views of the same image is the supervision signal for prediction. Probabilistic and variational models also learn prompt distributions that match the spreads of visual features (Liu et al., 2023; Derakhshani et al., 2022). Finally, MaPLe (Khattak et al., 2023) enhances the compatibility between the encoders of CLIP at different levels. However, these works only focus on learning a better prompt to improve the model's generalization ability. We think that the improvement from training the prompt alone is limited. Therefore, we propose a new fine-tuning method to handle the domain shift.

## 3 METHODOLOGY

Our method focuses on fine-tuning the image branch of a pre-trained multimodal CLIP for improved performance in domain generalization. Figure 3 presents the overall architecture of our proposed Im-Tuning framework. Unlike previous works that solely concentrate on optimizing prompts, Im-Tuning offers a pathway from the prompt to the image branch, allowing the model to learn more separable image features while optimizing the prompt.

### 3.1 PROBLEM SETUP OF DG

The DG problem involves $\mathcal{N}$ labelled source domains $\mathbf{S}^i = \left\{ \mathbf{x}_i^k, \mathbf{y}_i^k \right\}_{k=1}^{n_i} \sim P_{data}^{\mathbf{S}^i}, 1 \leq i \leq \mathcal{N}$, where $\mathbf{x}_i \in \mathbf{X}^i, \mathbf{y} \in \mathbf{Y}_i$, and $P_{data}^{\mathbf{S}^i}$ denote the input data, label, and the joint distribution for the label space, respectively. Furthermore, $P_{data}^{\mathbf{S}^i} \neq P_{data}^{\mathbf{S}^j}, \forall i, j \in \{1, 2, \cdots, \mathcal{N}\}$, showing that the source domains are diverse. We call the setting single-source DG if $\mathcal{N} = 1$, else it is known as multi-source DG. We aim to train a model $f : \mathbf{X} \rightarrow \mathbf{Y}$ given $\mathbf{S} = \left\{ \mathbf{S}^i \right\}_{i=1}^{\mathcal{N}}$, that can generalize to a new target domain $\mathbf{S}^{\mathcal{N}+1} = \left\{ \mathbf{x}_t^k, \mathbf{y}_t^k \right\}_{k=1}^{n_t}$ unseen during training with $\mathbf{x}_t \in \mathbf{X}^t$ and $\mathbf{y}_t \in \mathbf{Y}^t$ and $P_{data}^t$ as the target distribution different from the source distributions. We assume a closed-set setting where $\mathbf{Y} \cup \mathbf{Y}^t = \mathbf{Y} \cap \mathbf{Y}^t$. For the base to new class generalization setting, we have $\mathbf{Y} \cap \mathbf{Y}^t = \emptyset$.

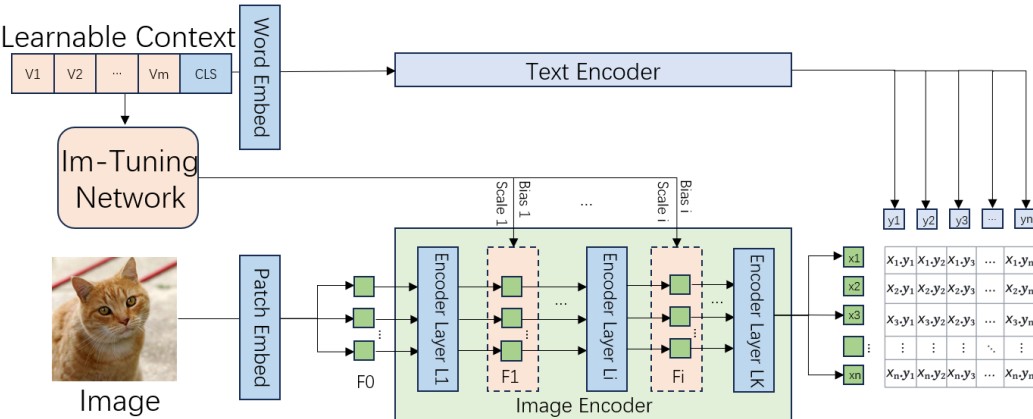

Figure 3: Overview of our proposed Im-Tuning framework in V-L models. Im-Tuning trains context prompts and allows prompts to learn scale and bias parameters through a linear network, which are applied to image features at different layers.

## 3.2 REVIEW OF COOP IN DG

Context Optimization aims to address the inefficiency challenge in prompt engineering for better adapting pre-trained vision-language models to downstream applications Zhou et al. (2022b). The key idea in Context Optimization is to model the context token as a continuous vector that can be end-to-end learned from data. Specifically, instead of utilizing "a photo of a" as the context, Context Optimization introduces M learnable contextual representation vectors, $\{\mathbf{v}_1, \mathbf{v}_2, \dots, \mathbf{v}_M\}$, each having an identical dimension to the word embeddings. The prompt for the $i$-th class, denoted as $\mathbf{t}_i$, now becomes $\mathbf{t}_i = \{\mathbf{v}_1, \mathbf{v}_2, \dots, \mathbf{v}_M, \mathbf{c}_i\}$ where $\mathbf{c}_i$ is the word embedding(s) for the class name. The contextual representation vectors are shared across all classes. The predictive probability is then

$$p(y \mid \mathbf{x}) = \frac{\exp\left(\cos\left(\mathbf{x}, g\left(\mathbf{t}_y\right)/\tau\right)\right)}{\sum_{i=1}^{K} \exp\left(\cos\left(\mathbf{x}, g\left(\mathbf{t}_i\right)/\tau\right)\right)}, \tag{1}$$

where $\cos(\cdot, \cdot)$ denotes cosine similarity , $\tau$ is a learned temperature parameter and $g(\cdot)$ denotes the frozen text encoder. The aims of Context Optimization is to learn better prompt.

## 3.3 IM-TUNING

As demonstrated in the introduction section (see Figure **??**), relying solely on prompt tuning is insufficient. Even the optimal prompts can barely improve the performance over a simple tuning strategy. However, image tuning can directly affect the images, making the resulting image features more separable. Thus, we emphasize the significance of image tuning for more effective optimization.

It is important to note that in the text branch, our settings are similar to CoOp (Zhou et al., 2022b) due to the absence of fine-tuning . Our objective is to amplify the distinctiveness of the image features by fine-tuning the image branch. Furthermore, the deepest layers of a visual encoder usually capture abstract object semantics for classification, but they miss local patterns such as edge orientations or local shapes (Zheng et al., 2016). Therefore, our approach focuses on fine-tuning different layers of the image encoder, which brings about stronger generalization and transferability compared to fine-tuning a single layer alone. Based on the feature sets from multiple levels and instance feature statistics, we construct a continuous prompt embedding space through linear mapping, allowing the learnable context to generate latent features in this embedding space, and further influence the multi-level visual features in the image encoder. In addition, we fine-tune the image feature maps in the first $J$ layers of the visual branch (where $J < K$, and $K$ represents the total number of layers in the visual branch, which is 12 in VIT-B/16. The analysis of the hyper-parameters $J$ is in appendix).

To perform image tuning, we first apply a linear mapping to the learnable prompt. The objective is to obtain a shared hidden layer feature based on the prompt. Similar to CoOp, the prompt given to the text encoder is designed with the following form

$$t = [\mathbf{V}]_1[\mathbf{V}]_2 \ldots [\mathbf{V}]_M[\mathbf{CLASS}], \tag{2}$$

where $[\mathbf{V}]_m (m \in \{1, \ldots, M\})$ is a vector with the same dimension as word embeddings (i.e., 512 for CLIP), and $M$ is a hyper-parameter specifying the number of context tokens.

$$L_f = \text{LinerProj}(t). \tag{3}$$

We generate scale and bias parameters for each transformer block of the image encoder up to a specific depth J by using different MLPs to process the latent features.

$$\begin{cases} \mathbf{scale}_i = \text{MLP}_{i-s}(L_f) \\ \mathbf{bias}_i = \text{MLP}_{i-b}(L_f) \end{cases}, \tag{4}$$

where each $\mathbf{scale}_i$ and $\mathbf{bias}_i$ $(i \in \{1, \ldots, J\})$ is a vector with the same dimension as the length of the token (i.e, 768 for ViT-B/16). We then apply the resulting scale and bias to each channel of the feature map output from the i-th layer.

$$\begin{cases} F_i = (\mathbf{scale}_i + 1)\mathcal{T}_i(F_{i-1}) + \mathbf{bias}_i & i = 1, 2, \cdots, J \\ F_i = \mathcal{T}_i(F_{i-1}) & i = J + 1, \cdots, K \end{cases}, \tag{5}$$

where $\mathcal{T}_i$ denotes the $i$-th layer of the image encoder. Next, we calculate the cosine similarity between the refined image features and text features, which determines the categorization of each image. We conduct experiments using two visual backbones, namely ResNet50 and ViT-B/16. In this section, we will elaborate on our method with ViT-B/16 as the visual backbone. We follow a similar method when using ResNet50 as the visual backbone, with the main distinction being that the scale and bias dimensions in ResNet50 are dependent on the output channel number of the $i$-th ResNet block. (e.g., the scale and bias dimensions are 256 after the first ResNet block in ResNet50, matching the channel number of the output feature map ).

During training, we update the context vectors along with parameters of $\text{MLP}_{i-s}$ and $\text{MLP}_{i-b}$ $(i \in \{1, \ldots, J\})$. In this work, these MLPs have a one-layer bottleneck structure (Linear-ReLU). We leave the exploration of more advanced designs for future work.

## 4 EXPERIMENT

Our approach is evaluated in the following problem settings: 1) few-shot learning (Section 4.2) 2) generalization from base to new classes within a dataset (Section 4.3); 3) cross-dataset transfer (Section 4.4); 4) domain generalization (Section 4.5). All models used in our experiments are based on CLIP. Before discussing the results, we provide details of the experimental setup below.

### 4.1 EXPERIMENTAL PROTOCOL

**Datasets.** For the first three settings, i.e., few-shot learning, base-to-new generalization, and cross-dataset transfer, we use the 11 image recognition datasets as in Zhou et al. (2022b), which cover a diverse set of recognition tasks. Specifically, the benchmark includes ImageNet (Deng et al., 2009) and Caltech101 (Fei-Fei et al., 2004) for classification on generic objects; OxfordPets (Parkhi et al., 2012), StanfordCars (Krause et al., 2013),Flowers102 (Krause et al., 2013), Food101 (Bossard et al., 2014) and FGVCAircraft (Maji et al., 2013) for fine-grained classification; SUN397 (Xiao et al., 2010) for scene recognition; UCF101 (Soomro et al., 2012) for action recognition; DTD (Cimpoi et al., 2014) for texture classification; and finally EuroSAT (Helber et al., 2019) for satellite imagery recognition. For domain generalization experiments, We evaluate over 3 benchmark datasets for multi-source DG. They are: (1) Office-Home (Venkateswara et al., 2017) (2) PACS (Li et al., 2017) (3) DomainNet (Peng et al., 2019) . We conduct experiments on 4 domains of DomainNet, i.e., Clipart, Painting, Real, and Sketch.

**Hyper-parameters and model selection.** Our codes are built on top of the open-source code of CLIP and CoOp. We use the SGD optimizer (Bottou, 2012) to train the model with a learning rate

of 2e-3 and betas values (0.9, 0.999), which is decayed by the cosine annealing rule. Regarding the prompt configuration, we set the prompt length to 4 in all experimental results. In the baseline network with ResNet50 (He et al., 2016), we use prompt embedding to generate scale and bias parameters that influence the image feature maps obtained from the first two ResNet blocks. In the baseline network using ViT-B/16 (Dosovitskiy et al., 2020), we use prompt embedding to modify the feature maps obtained from the first $J$ layers of the transformer layer. In our experiments, we set $J$ to 9, and further discussion of the impact on length of context, number of layers, and model selection can be found in the ablation study. Finally, we set the batch size to 16 and train the model for 10 epochs. For all our training, we utilize only two RTX3090 GPUs. We report the average top-1 classification performance on each dataset based on three different executions of our method.

**Baseline.** For few-shot learning, we compare Im-Tuning with CLIP zero-shot, and recent prompt learning works including CoOp (Zhu et al., 2022). For generalization form base to new and cross dataset transfer, we compare Im-Tuning with CLIP zero-shot, CoOp (Zhu et al., 2022) and Co-CoOp (Zhou et al., 2022a) . For domain generalization experiments, We consider various methods for comparison, including M3SDA (Peng et al., 2019), which is a traditional method that uses moment matching to minimize the discrepancy between the source domain and target domain. Additionally, a more basic method is zero-shot CLIP with the prompt "A Photo of a [CLS]". Furthermore, we choose to compare our method with several existing prompt tuning methods, e.g., CoOp, CoCoOp, and DPL Zhang et al. (2021a). Finally, we also compare our method with a fine-tune method, e.g., CLIP-Adapter Gao et al. (2021), which fine-tunes adapters in both visual and language branches.

## 4.2 FEW-SHOT LEARNING

We follow the few-shot evaluation protocol adopted in CLIP, using 1, 2, 4, and 16 shots for training respectively, and deploying models in the full test sets. The maximum epoch is set to 200 for 16 shots, 100 for 4/2 shots, and 50 for 1 shot (except for ImageNet where the maximum epoch is fixed to 50). The context length in CoOp is set to 4. We report the average results over three runs with different random seeds. Zero-shot CLIP is based on hand-crafted prompts, which is adopted in CLIP.

In general, our method outperforms CoOp and zero-shot CLIP on all 11 datasets, demonstrating strong few-shot learning capabilities. Among the four settings of 11 datasets in total, Im-Tuning achieves an average improvement of 6.58% over CoOp.

## 4.3 GENERALIZATION FROM BASE TO NEW CLASSES

We address the weak generalization problem of CoOp by proposing Im-Tuning, which can learn from a subset of classes and generalize to unseen classes. We conduct experiments on 11 recognition datasets, where we divide the classes into base and novel sets. We train the learning-based models, i.e., CoOp, CoCoOp and Im-Tuning, on the base classes and evaluate them on both the base and novel classes separately. Table 1 shows the results of this setting, where we compare Im-Tuning with CLIP zero-shot, and other prompt learning methods such as CoOp and CoCoOp. For CLIP, we use hand-crafted prompts that are tailored for each dataset. Im-Tuning outperforms CoCoOp on both the base and novel classes on all 11 datasets, except for a slight decrease on the base classes of Caltech101 and Food101. Im-Tuning also achieves better generalization to novel classes on 8 out of 11 datasets, compared to CoCoOp. Im-Tuning improves the overall accuracy on all 11 datasets from 71.69% to 73.11%, and obtains an average gain of 1.57% over CoCoOp. These results demonstrate the effectiveness of Im-Tuning in cross-class generalization.

## 4.4 CROSS-DATASET TRANSFER

To evaluate the cross-dataset generalization ability of Im-Tuning, which is a more challenging problem than within-dataset generalization as the data distribution and task domain may vary significantly, we compare it with CLIP , CoOp and CoCoOp on 10 different datasets using ImageNet as the source dataset. We use all 1,000 classes of ImageNet to learn the context for prompt learning methods. Table 2 shows the results of this experiment. On ImageNet, CoOp achieves the highest accuracy, but on the other datasets, Im-Tuning consistently outperforms CoOp and CoCoOp by a

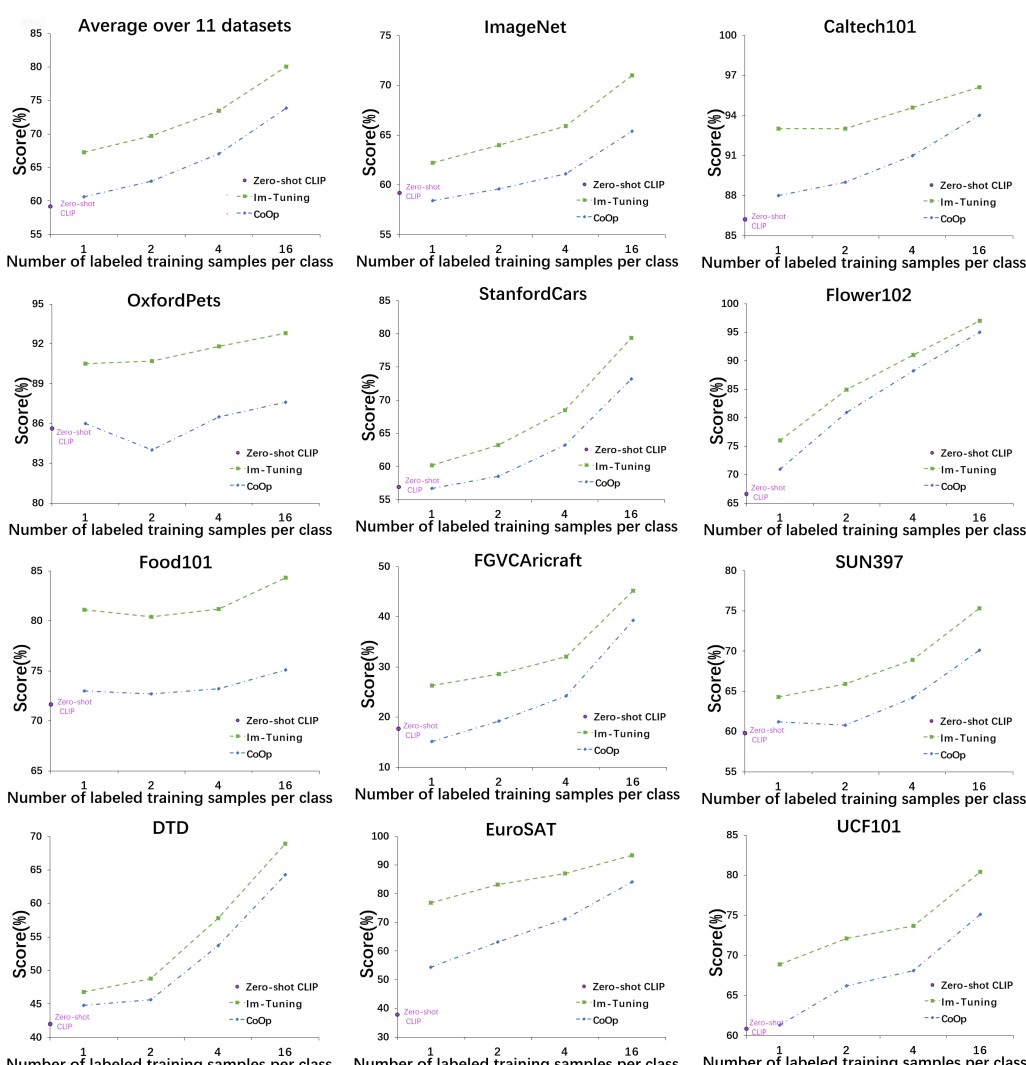

Figure 4: Main results of few-shot learning on the 11 datasets.

large margin. The high accuracy on Caltech101 and OxfordPets can be attributed to the fact that ImageNet contains many object and dog classes that are relevant to these datasets.

## 4.5 DOMAIN GENERALIZATION

We present experimental results on the DomainNet, Office-Home, and PACS datasets using two visual backbones, ResNet50 and ViT-B/16 in Table 3. As expected, ViT-B/16, being a more powerful backbone, outperforms ResNet50 in terms of performance. We observed that traditional algorithms that do not utilize large foundation models, such as M3SDA, tend to have lower performance across all three datasets (averaging around 10% lower compared to methods incorporating CLIP). Additionally, the zero-shot CLIP method performs relatively poorly in domain generalization tasks, achieving only 78.2% and 68.1% accuracy on the Office-Home and DomainNet datasets, respectively. These results highlight a significant gap compared to prompt tuning methods like COOP, emphasizing the effectiveness of prompt tuning in enhancing model generalization abilities.

Our method consistently surpasses the performance of existing methods across various datasets and visual backbones, demonstrating its effectiveness and robustness. Among the methods with ViT-B/16 as the visual backbone, our method achieves accuracy of 75.25%, 85.7%, and 98.0% on the DomainNet, Office-Home, and PACS datasets, respectively. On average, our method shows an improvement of over 1% compared to other methods.

Table 1: Comparison with state-of-the-art methods on base-to-novel generalization. Im-Tuning learns demonstrates strong comprehensive results over existing methods on 11 recognition datasets.

(a) **Average over 11 datasets**

| | Base | New | H |
|---|---|---|---|
| CLIP | 69.34 | 74.22 | 71.70 |
| CoOp | 82.69 | 63.22 | 71.66 |
| CoCoOp | 80.47 | 71.69 | 75.83 |
| Im-Tuning | 82.23 | 73.11 | **77.40** |

(b) ImageNet

| | Base | New | H |
|---|---|---|---|
| CLIP | 72.43 | 68.14 | 70.22 |
| CoOp | 76.47 | 67.88 | 71.92 |
| CoCoOp | 75.98 | 70.43 | 73.10 |
| Im-Tuning | 76.23 | 70.11 | 73.04 |

(c) Caltech101

| | Base | New | H |
|---|---|---|---|
| CLIP | 96.84 | 94.00 | 95.40 |
| CoOp | 98.00 | 89.81 | 93.73 |
| CoCoOp | 97.96 | 93.81 | 95.84 |
| Im-Tuning | 97.92 | **94.44** | **96.15** |

(d) OxfordPets

| | Base | New | H |
|---|---|---|---|
| CLIP | 91.17 | 97.26 | 94.12 |
| CoOp | 93.67 | 95.29 | 94.47 |
| CoCoOp | 95.20 | **97.76** | 96.43 |
| Im-Tuning | 95.44 | 97.47 | **96.44** |

(e) StranfordCars

| | Base | New | H |
|---|---|---|---|
| CLIP | 63.37 | 74.89 | 68.65 |
| CoOp | 78.12 | 60.40 | 68.13 |
| CoCoOp | 70.49 | 73.59 | 72.01 |
| Im-Tuning | 73.38 | 74.23 | **73.80** |

(f) Flower102

| | Base | New | H |
|---|---|---|---|
| CLIP | 72.08 | **77.80** | 74.83 |
| CoOp | 97.60 | 59.67 | 74.06 |
| CoCoOp | 94.87 | 71.75 | 81.71 |
| Im-Tuning | 95.83 | 72.29 | **82.41** |

(g) Food101

| | Base | New | H |
|---|---|---|---|
| CLIP | 90.10 | 91.22 | 90.66 |
| CoOp | 88.33 | 82.26 | 85.19 |
| CoCoOp | **90.70** | 91.29 | **90.99** |
| Im-Tuning | 90.23 | 91.17 | 90.70 |

(h) FGVCAircraft

| | Base | New | H |
|---|---|---|---|
| CLIP | 27.19 | **36.29** | 31.09 |
| CoOp | 40.44 | 22.30 | 28.75 |
| CoCoOp | 33.41 | 23.71 | 27.74 |
| Im-Tuning | 37.17 | 34.49 | **35.78** |

(i) SUN397

| | Base | New | H |
|---|---|---|---|
| CLIP | 69.36 | 75.35 | 72.23 |
| CoOp | 80.60 | 65.89 | 72.51 |
| CoCoOp | 79.74 | 76.86 | 78.27 |
| Im-Tuning | **81.10** | 76.82 | **78.90** |

(j) DTD

| | Base | New | H |
|---|---|---|---|
| CLIP | 53.24 | **59.90** | 56.37 |
| CoOp | 79.44 | 41.18 | 54.24 |
| CoCoOp | 77.01 | 56.00 | 64.85 |
| Im-Tuning | **79.52** | 56.79 | **66.26** |

(k) EuroSAT

| | Base | New | H |
|---|---|---|---|
| CLIP | 56.48 | **64.05** | 60.03 |
| CoOp | 92.19 | 54.74 | 68.69 |
| CoCoOp | 87.49 | 60.04 | 71.21 |
| Im-Tuning | 93.34 | 60.22 | **73.21** |

(l) UCF101

| | Base | New | H |
|---|---|---|---|
| CLIP | 70.53 | **77.50** | 73.85 |
| CoOp | **84.69** | 56.05 | 67.46 |
| CoCoOp | 82.33 | 73.45 | 77.64 |
| Im-Tuning | 84.41 | 76.23 | **80.11** |

Table 2: Comparison of Im-Tuning with existing approaches on cross-dataset evaluation. Overall, Im-Tuning achieves competitive performance providing the highest average accuracy, indicating better generalization.

| | Source | Target | | | | | | | | | | |
|---|---|---|---|---|---|---|---|---|---|---|---|---|
| | ImageNet | Caltech101 | OxfordPets | StanfordCars | Flowers102 | Food101 | Aircraft | SUN397 | DTD | EuroSAT | UCF101 | Average |
| CoOp | **71.51** | 93.70 | 89.14 | 64.51 | 68.71 | 85.30 | 18.47 | 64.15 | 41.92 | 46.39 | 66.55 | 63.88 |
| Co-CoOp | 71.02 | 94.43 | 90.14 | **65.32** | 71.88 | 86.06 | 22.94 | 67.36 | 45.73 | 45.37 | 68.21 | 65.74 |
| Im-Tuning | 71.21 | **94.45** | **90.37** | 64.86 | **72.25** | **86.17** | **23.62** | **67.72** | **44.62** | **46.64** | **69.41** | **66.48** |

## 4.6 ABLATION STUDY

**Length of context.** In prompt tuning, the length of context is a crucial variable, and its optimal

Table 3: Comparison of our proposed method with the state-of-the-art methods on PACS, Office-Home, and DomainNet datasets for multi-source DG.

| Backbone | Method | PACS | Office-Home | DomainNet | Avg |
|---|---|---|---|---|---|
| RN50 | M3SDA | 85.1 ± 1.1 | 67.2 ± 1.2 | 56.1 ± 1.1 | 69.5 |
| | ZS-CLIP | 90.2 ± 0.5 | 70.1 ± 0.4 | 58.2 ± 0.3 | 72.8 |
| | CoOp | 92.3 ± 0.5 | 74.8 ± 0.4 | 62.4 ± 0.4 | 76.5 |
| CLIP | CoCoOp | 92.1 ± 0.4 | 75.1 ± 0.4 | 62.6 ± 0.4 | 76.6 |
| RN50 | DPL | 91.9 ± 0.3 | 75.4 ± 0.5 | 62.7 ± 0.4 | 76.7 |
| | CLIP-Adapt | 92.1 ± 0.3 | 75.1 ± 0.4 | 62.6 ± 0.2 | 76.6 |
| | Im-Tuning | **92.8 ± 0.3** | **75.8 ± 0.4** | **63.1 ± 0.3** | **77.2** |
| | ZS-CLIP | 95.9 ± 0.2 | 80.6 ± 0.3 | 68.2 ± 0.1 | 81.6 |
| | CoOp | 97.0 ± 0.1 | 84.2 ± 0.2 | 74.1 ± 0.2 | 85.1 |
| CLIP | CoCoOp | 96.7 ± 0.2 | 84.3 ± 0.1 | 73.9 ± 0.4 | 85.0 |
| ViT-B/16 | DPL | 97.1 ± 0.3 | 84.4 ± 0.2 | 74.3 ± 0.3 | 85.3 |
| | CLIP-Adapt | 96.3 ± 0.1 | 83.6 ± 0.2 | 74.2 ± 0.2 | 84.7 |
| | Im-Tuning | **97.8 ± 0.1** | **85.6 ± 0.2** | **75.2 ± 0.2** | **86.2** |

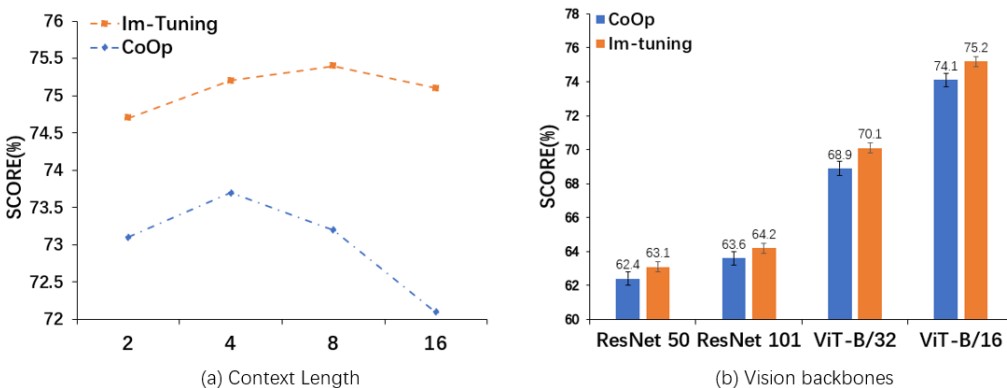

Figure 5: Investigations on Im-Tuning's context length and various vision backbones.

setting has been widely discussed in relevant literature. As is mentioned by the authors of CoOp in their paper, continuously increasing the length of context for domain generalization can actually lead to worse results. This may be due to overfitting caused by a smaller number of learnable parameters. We also observe this phenomenon in CoOp. However, in our method, the introduction of more learnable parameters has made our method less sensitive to the length of context. We do not observe significant overfitting when increasing the length of context. The specific data regarding this observation are presented in Figure 5(a).

**Vision backbone.** Many proposed DG methods are evaluated using the standard ResNet backbones. However, more and more large models are being studied, and their validity is being experimentally demonstrated. Therefore, we report the performance of ResNet50, ResNet101, and several variations of Vision Transformer (ViT). Figure 5(b) shows the results on DomainNet using these backbones. The results are expected: the more advanced the backbone, the better the performance. Our Im-Tuning outperforms CoOp across all architectures, especially in the ViT backbones.

**Separable image feature.** To investigate whether our Im-Tuning method can produce better image features, we compare the image features of Im-Tuning and CoOp In Figure 6 (see the appendix). The image features of CLIP, CoOp, and Co-CoOp are the same as they do not learn prompts in the vision branch. The visualization shows that image features of Im-Tuning are more separable, suggesting that tuning the image encoder improves the adaptation of CLIP.

## 5 CONCLUSION, LIMITATIONS AND FUTURE WORK

Large pre-trained vision-language models have shown their powerful capabilities in various downstream tasks. There is also extensive research in the domain generalization field on how to leverage these models to overcome domain shift. Existing methods mainly rely on prompt tuning to address domain shift, but they have limitations. In our paper, we propose a lightweight network that allows the visual backbone of large models to participate in gradient backpropagation, thereby enhancing their performance in domain generalization tasks.

While our method achieves promising results, it still lacks interpretability, and it is sensitive to noisy labels. Performance can significantly decrease when facing poorly annotated data, which is a common challenge for many methods based on large Vision-Language models.

Im-Tuning is a simple yet effective method that enables the visual backbone of large pre-trained vision-language models to fine-tune on novel domains. This opens up many avenues for future research, such as investigating the cross-dataset transferability and test-time adaptability of Im-Tuning.

In summary, our proposed method is simple, effective, and exhibits good scalability. We hope that the findings of this research can pave the way for the application of various large Vision-Language models in domain generalization tasks. It remains an interesting and practical research topic.

## ETHIC STATEMENT

This paper does not raise any ethical concerns. This study does not involve any human subjectspractices to data set releases, potentially harmful insights, methodologies and applications, potentialconflicts of interest and sponsorship, discrimination/bias/fairness concerns, privacy and securityissues.legal compliance,and research integrity issues.

## REPRODUCIBILITY STATEMENT

To make all experiments reproducible, we have listed all detailed hyper-parameters of each experiment. Due to privacy concerns, we will upload the anonymous link of source codes andinstructions during the discussion phase to make it only visible to reviewers.

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

## A APPENDIX

### A.1 DATASET DETAILS

We provide our experimental results on four variants of ImageNet in Table 5 and we also provide detailed statistical information for these datasets in Table4: The source dataset is ImageNet. The target datasets are ImageNetV2, ImageNet-Sketch, ImageNet-A, and ImageNet-R, all of which have compatible class names with ImageNet. ImageNetV2 is a reproduced test set using different sources while following ImageNet's data collection process. ImageNet-Sketch contains sketch images belonging to the same 1,000 ImageNet classes. Both ImageNet-A and -R contain 200 classes derived from a subset of ImageNet's 1,000 classes.

### A.2 EXPLANATION OF SYMBOLS

We have compiled a comprehensive list of special symbols used in the formulas throughout the article and provided detailed explanations for each of them in table 6.

Table 4: Statistical information for four variants of ImageNet.

| Dataset | Classes | Train | Val | Test | Hand-crafted prompt |
|---|---|---|---|---|---|
| ImageNetV2 | 1000 | N/A | N/A | 10000 | |
| ImageNet-Sketch | 1000 | N/A | N/A | 50889 | "a photo of a [CLASS]." |
| ImageNet-A | 200 | N/A | N/A | 7500 | |
| ImageNet-R | 200 | N/A | N/A | 30000 | |

Table 5: Comparison of Im-Tuning with existing approaches in domain generalization setting.

| | Source | Target | | | |
|---|---|---|---|---|---|
| | ImageNet | ImageNetV2 | ImageNet-S | ImageNet-A | ImageNet-R |
| CLIP | 66.73 | 60.83 | 46.15 | 47.77 | 73.96 |
| CoOp | **71.51** | 64.20 | 47.99 | 49.71 | 75.21 |
| Co-CoOp | 71.02 | 64.07 | 48.75 | **50.63** | 76.18 |
| Im-tuning | 71.21 | **64.31** | **49.12** | 50.22 | **76.67** |

## A.3 ABLATION ANALYSIS

**Analysis of upper bound.** Multi-modal large-scale models like CLIP have shown strong domain generalization capabilities in the field of domain generalization. For instance, zero-shot CLIP achieves an accuracy of 78% on the Office-Home dataset without any training, outperforming traditional domain generalization methods designed for the Office-Home dataset. Prompt tuning methods based on CLIP have enhanced the accuracy of domain generalization. However, we have also seen some unusual phenomena. In domain generalization tasks, prompt tuning methods sometimes have lower training accuracy than testing accuracy in some domains. For instance, using CoOp for domain generalization on the DomainNet dataset, training on the other three domains, and generalizing to the real domain, the training accuracy is only 71%, while the testing accuracy is 85%. Such a phenomenon is unlikely to occur in traditional domain generalization methods.

We attribute this phenomenon partly to the inherent bias of large models like CLIP towards different domains. This also suggests that current prompt tuning methods may have some limitations. To measure these limitations, we introduce the concept of the upper bound. The upper bound is the prompt trained on all the data from each domain of the multi-source DG datasets with enough epochs until convergence (in this study, we train for 200 epochs). In Table 7, we compare the upper bounds of CLIP fine-tuning methods, various prompt tuning methods, and our method on the DomainNet dataset. Table 7 shows that the upper bound of CLIP fine-tuning methods is relatively low, only 76.2%. Prompt tuning methods have an advantage in this aspect, with CoOp and CoCoOp achieving higher upper bounds. CoCoOp has a slightly higher upper bound due to the linear networks that allow image features to influence the prompt. However, both CoOp and CoCoOp are prompt tuning methods, with CoCoOp's upper bound being only 81%. This implies that training on prompts alone is not enough, as even the optimal prompt with known testing data can only reach 81% accuracy. In contrast, our method can achieve an upper bound of 89.6%. This indicates that our method has more room for improvement than prompt tuning methods.

**Initialization.** We experiment on the DomainNet and Office-Home datasets, using "a photo of a" as the initialization vector to compare with the random initialization in our method (random initialization means being randomly initialized from a zero-mean Gaussian distribution with a standard deviation of 0.02). The experimental results in Table 8 indicate that better initialization does not necessarily lead to performance improvement, suggesting that there is no need to pay excessive attention to the design of the prompt. The model can learn a good prompt in the process of backpropagation.

**Number of layers.** To observe the impact of the number of layers on the experimental results, we use ViT B/16 as the backbone network. We investigate the effects of modifying the scale and bias for different transformer layers, with a context length of $4/16$. The results are recorded in the Table 9. From the results, it can be observed that modifying transformer layers 1-9 yields the best average results for the domain generalization task. Therefore, we adopt this configuration in our experiments.

Table 6: Description of key symbols in this paper

| Symbol | Description |
|---|---|
| $\mathcal{S}^i = \{x_i^k, y_i^k\}_{k=1}^{n_i}$ | is the definition of source domain in DG problem |
| $P_{data}^{\mathcal{S}^i}$ | is the joint distribution concerning the data and the label space of source domain |
| $\mathcal{S}^{\mathcal{N}+1} = \{x_t^k, y_t^k\}_{k=1}^{n_t}$ | is the definition of target domain in DG problem |
| $P_{data}^t$ | is the joint distribution concerning the data and the label space of target domain |
| $g(\cdot)$ | is the text encoder in CLIP model |
| t | is the prompt given to the text encoder in our method |
| $L_f$ | is the hidden layer features obtained through linear mapping of the learnable prompt. |
| J | is the number of layers fine-tuned using our method in the image encoder. |
| $scale_i, bias_i$ | is the parameters applied to the image encoder layer i ($i \in \{1, \ldots, J\}$) |
| $F_i$ | is the output of the image encoder layer i ($i \in \{1, \ldots, J\}$) |

Table 7: Comparison of our proposed method with CLIP- based state-of-the-art methods for upper bound on DomainNet and Office-Home. CLIP Liner is a model obtained by training an additional linear classifier on top of CLIP

| Method | CLIP Liner | CoOp | CoCoOp | Im-Tuning |
|---|---|---|---|---|
| upper bound on DomainNet | 76.2 | 79.3 | 81.0 | **88.6** |
| upper bound on Office-Home | 86.2 | 88.1 | 89.2 | **94.5** |

Table 8: Investigations on initialization of our method. We report the average top-1 classification performance for domain generalization on DomainNet and Office-Home.

| Initialization | DomainNet | Office-Home |
|---|---|---|
| random initialization | 75.2 | 85.6 |
| "a photo of a" | 75.2 | 85.5 |

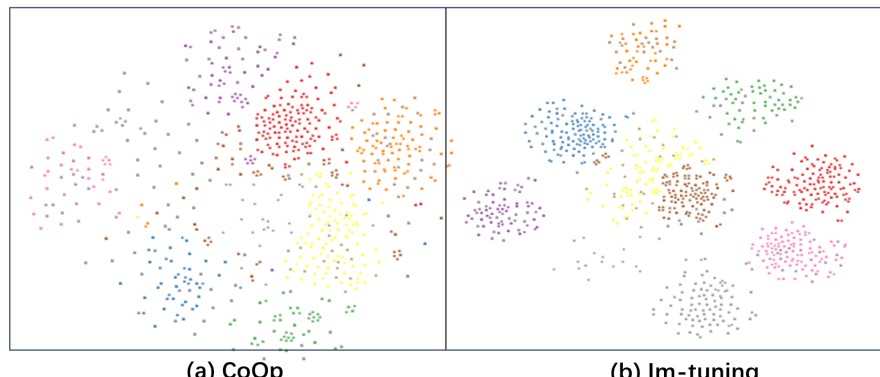

(a) CoOp         (b) Im-tuning

Figure 6: t-SNE plots of image features in prompt tuning method CoOp, and our Im-Tuning on Office-Home dataset. Im-Tuning shows better separability.

Table 9: Ablation analysis of the number of layers in Im-Tuning using ViT-B/16 backbone (In%)

| Baselines | DomainNet |
|---|---|
| context length=4,transformer layer 1-3 | 74.3 |
| context length=4,transformer layer 1-6 | 74.8 |
| context length=4,transformer layer 1-9 | **75.2** |
| context length=4,transformer layer 1-12 | 74.9 |
| context length=16,transformer layer 1-3 | 74.2 |
| context length=16,transformer layer 1-6 | 74.6 |
| context length=16,transformer layer 1-9 | **75.0** |
| context length=16,transformer layer 1-12 | 74.8 |

