# OpenReview forum: "Prompt Tuning Is All We Need?"
_ICLR.cc/2024/Conference — Submitted to ICLR 2024_

### Official Review · Reviewer_Xqq5 · 2023-10-31

**Soundness:** 3 good
**Presentation:** 3 good
**Contribution:** 3 good
**Rating:** 3
**Confidence:** 5

**Summary:**

The authors present in-tuning that fine-tunes the CLIP text and visual backbones on downstream tasks. The authors add learnable tokens and the scale and bias parameters for the transformer block of the CLIP image encoder. The authors provide a detailed analysis on 11 image recognition and 3 domain generalization benchmarks.

**Strengths:**

1. The idea of this paper is easy to understand and easy to follow

**Weaknesses:**

- 1. Title Clarity. The current title appears too broad and potentially misleading readers. Initially, one might assume the paper provides an analysis of the pros and cons of prompt tuning across various tasks. However, the main content focuses on the present new method of tuning the CLIP image encoder, which is not reflected in the title.

- 2. Lack of novelty. Some previous work, such as MaPLe (Khattak et al., 2023) and CLIP-Adapter Gao et al. (2021) have already delved into tuning the CLIP image encoder and making the CLIP image features separable.

- 3. Missing baselines. The authors only compare with CoOp/CoCoOp in Table 1 and Table 2, which are not enough. CoOp/CoCoOp are published two years ago and not the SOTA method. A series of methods mentioned in the related work section are suggested to compare with, such as ProGrad (Zhu et al., 2022), Test-time prompt tuning (Shu et al. (2022) and MaPLe (Khattak et al., 2023). They also provide the results on the base-to-new evaluation and cross-dataset transfer settings. Some recent prompt tuning approaches that are not listed in the related work section are also suggested to compare with, such as

  - LASP Language Aware Soft Prompting for Vision-Language Models, CVPR 2023

- 4. Efficiency concerns. The introduction of the in-tuning network adds more parameters, possibly affecting the training and inference times and raising the efficiency concern. The authors may consider adding an efficiency analysis and providing a comprehensive view of the trade-offs involved.

**Questions:**

1. What's the meaning of L_{f} in Eq (3)?

---

> ### Author Response · Authors · 2023-11-22
>
> Thank you for reading and appreciating our work! We address your questions and concerns below.
> Q1:Title Clarity. The current title appears too broad and potentially misleading readers. Initially, one might assume the paper provides an analysis of the pros and cons of prompt tuning across various tasks. However, the main content focuses on the present new method of tuning the CLIP image encoder, which is not reflected in the title.
>
> A1:Thank you for your feedback on the Title Clarity. We primarily intended to convey our research findings in the context of domain generalization tasks. We appreciate your suggestion, and we will make sure to revise our title accordingly.
>
> Q2:Missing baselines. The authors only compare with CoOp/CoCoOp in Table 1 and Table 2, which are not enough.CoOp/CoCoOp are published two years ago and not the SOTA method. A series of methods mentioned in the related work section are suggested to compare with, such as ProGrad (Zhu et al., 2022), Test-time prompt tuning (Shu et al. (2022) and MaPLe (Khattak et al., 2023).
>
> A2:Thank you for your feedback regarding the missing baselines. Our Im-Tuning method outperforms MaPLe and ProGrad in domain generalization tasks. Specifically, we conducted experiments on three DG datasets, namely PACS, Office-Home, and DomainNet. Our method achieved an average accuracy of 87.2%, while MaPLe achieved an average accuracy of 86.7%, and ProGrad achieved an average accuracy of only 85.4%. Additionally, our method is applicable to networks with ResNet as the backbone, further improving domain generalization performance, which is not achievable by MaPLe.

---

### Official Review · Reviewer_PxvM · 2023-10-31

**Soundness:** 2 fair
**Presentation:** 1 poor
**Contribution:** 2 fair
**Rating:** 3
**Confidence:** 5

**Summary:**

Recent advancements in pre-trained vision-language models, such as CLIP, have shown impressive success in domain generalization (DG) by fine-tuning prompts. One promising approach to enhance DG is prompt learning, which aims to design or learn more effective prompts. The underlying idea is that a more sophisticated prompt learning method can lead to better generalization performance. To investigate the impact of prompt learning on DG, comprehensive experiments were conducted on DG benchmarks.

Surprisingly, the experiments yielded a pessimistic finding. It was discovered that simply tuning prompts using training sets achieved comparable performance to using test sets. In other words, even with optimal prompts, significant performance improvement compared to a simple tuning strategy was difficult to achieve. This observation was attributed to the non-separability of features extracted by the image encoder.

To address this limitation, the researchers proposed a method called Im-Tuning, which focuses on tuning the image encoder to generate more separable image features. Extensive experiments were conducted on multiple DG benchmarks to evaluate the effectiveness of Im-Tuning. The results consistently demonstrated that Im-Tuning outperformed the state-of-the-art methods in DG tasks.

In summary, while previous work emphasized the importance of prompt learning for DG, this research revealed that prompt tuning alone is insufficient to achieve significant performance gains. Instead, the proposed Im-Tuning method, which focuses on enhancing the separability of image features through image encoder tuning, proved to be more effective in improving DG performance across various benchmarks.

**Strengths:**

1)  This work presents a seemingly new opinion: it challenges the necessity of prompt tuning, and shows that an optimal prompt provides limited performance gain compared to a simple prompt. This can be attributed
to the separability of image features, which suggests a promising direction for promoting
domain generalization with clip models.

2) The designed method is simple yet effective and meets intuition.

**Weaknesses:**

1) One main work [1] to compare in experiments is missing, as it focuses on multi-modal tuning. To convince the conclusion that tuning prompt and image enocder is better than multi-modal prompt tuning, more empirical results are required.

2) The presentation is not clear. The abstract is hard to understand.

3) Some writing errors, like "As demonstrated in the introduction section (see Figure ??)" in Sec 3.3

[1] MaPLe: Multi-modal Prompt Learning. CVPR23

**Questions:**

Please see weakness.

The design of modulating image features based on the prompts,  should be studied more by ablation.

---

> ### Author Response · Authors · 2023-11-22
>
> Thank you for reading and appreciating our work! We address your questions and concerns below.
> Q1:One main work to compare in experiments is missing, as it focuses on multi-modal tuning. To convince the conclusion that tuning prompt and image enocder is better than multi-modal prompt tuning, more empirical results are required.
>
> A1：Thank you for providing the baseline. We conducted additional experiments on three domain generalization datasets, and our results outperform MaPLe. On PACS, Office-Home, and DomainNet, MaPLe achieves an average accuracy of 86.7%, while our Im-Tuning achieves 87.2%. Furthermore, our experiments provide insights into methods like MaPLe that focus on multi-modal tuning. These methods are effective because they improve the representation of image features, making them more separable. We attempted to remove the image branch in MaPLe and observed a significant decrease in performance, which validates our viewpoint.

---

> > ### Comment · Reviewer_PxvM · 2023-11-22
> > **A table-like figure could be more helpful**
> >
> > Thanks for your response. As you acknowledged, Maple is a critical SOTA that should be compared, so a table-like result representation would be more helpful.

---

### Official Review · Reviewer_W47K · 2023-10-31

**Soundness:** 2 fair
**Presentation:** 3 good
**Contribution:** 3 good
**Rating:** 6
**Confidence:** 4

**Summary:**

This paper investigates methods for domain generalization using pretrained vision-language models. The authors introduce a novel image encoder tuning method called Im-Tuning, aimed at enhancing the separability of image features by adjusting the parameters of the image encoder. The empirical results demonstrate that the Im-Tuning method outperforms existing approaches across multiple domain generalization benchmarks.

**Strengths:**

1. The paper addresses an important research question regarding the effectiveness of prompt tuning in domain generalization for vision-language models.

2. The authors propose a new method, Im-Tuning, which improves the separability of image features by adjusting the image encoder, thereby enhancing domain generalization performance.

3. Extensive experiments on multiple domain generalization benchmarks validate the effectiveness of the Im-Tuning method.

4. The paper is well-structured, provides a thorough review of relevant literature, and presents a comparison of the proposed method with existing approaches.

**Weaknesses:**

1. The paper provides a rather brief description of experimental details, which lacks depth and may hinder reproducibility.

2. The clarity of the argument could be improved, as some parts are not expressed clearly enough, making it challenging for readers to understand.

3.  The interpretability of the method is poor, requiring further explanation on why adjusting the image encoder enhances the separability of image features.

4. The paper contains some formatting errors, such as in Section 3.3, the first sentence ("see Figure ??").

**Questions:**

1. Could the authors provide more detailed experimental settings and implementation details for reproducibility?

2. Could further explanation be provided on why adjusting the image encoder enhances the separability of image features?

3. Have other domain generalization tasks been considered, such as zero-shot learning or domain adaptation?

---

> ### Author Response · Authors · 2023-11-22
>
> Thank you for reading and appreciating our work! We address your questions and concerns below.
> Q1:Could the authors provide more detailed experimental settings and implementation details for reproducibility?
>
> A1:Thank you for raising this question. We apologize for any confusion caused.We use the SGD optimizer to train the model with a learning rate of 2e-3 and betas values (0.9, 0.999), which is decayed by the cosine annealing rule. we actually employ a similar cross-entropy loss function as used in CoOp to train the entire model.
>
> Q2:Could further explanation be provided on why adjusting the image encoder enhances the separability of image features?
>
> A2:Thank you for your inquiry. Our approach of adjusting the image encoder draws inspiration from traditional domain generalization and domain adaptation techniques that achieve improved performance by modifying the batch normalization layers of the model, as mentioned in references [1] and [2]. Building upon this idea, we propose to enhance the model's performance by altering the statistics of image features, i.e., scale and bias in our paper. This adjustment helps to enhance the separability of image features and improve the overall performance of the model.
>
> [1]Domain-Specific Batch Normalization for Unsupervised Domain Adaptation
>
> [2]Improving robustness against common corruptions by covariate shift adaptation

---

### Official Review · Reviewer_wRsx · 2023-11-01

**Soundness:** 2 fair
**Presentation:** 3 good
**Contribution:** 2 fair
**Rating:** 5
**Confidence:** 4

**Summary:**

This paper proposed a method to improve the domain generalization ability of the CLIP model. Specifically, they proposes image encoder tuning, where the learnable contextual representation vectors also control the scale and bias of the image encoder layers. The experimental results performed on few-shot learning, generalization from base to new classes, and cross-dataset transfer, domain generalization results showing the effectiveness of the proposed method.

**Strengths:**

- The paper is well-written and easy to understand
- The proposed method is simple and effective
- Extensive experiments showing the effectiveness of the proposed method.

**Weaknesses:**

- Better to provide detailed analysis about why image features of a pre-trained CLIP model is less separable. Are they related to training data or loss functions? Do you observed similar phenomenons in other CLIP-like models?

**Questions:**

I'm thinking about the scope of this paper. The proposed method is more like an incremental improvement over a pre-trained CLIP model. It is unknown if the drawback mentioned in the paper is only stands for this model checkpoint, for the training dataset, or for all CLIP-style models. It is interesting to see more analysis and comparisons with other multimodality models

---

> ### Author Response · Authors · 2023-11-22
>
> Thank you for reading and appreciating our work! We address your questions and concerns below.
> Q1:The proposed method is more like an incremental improvement over a pre-trained CLIP model. It is unknown if the drawback mentioned in the paper is only stands for this model checkpoint, for the training dataset, or for all CLIP-style models. It is interesting to see more analysis and comparisons with other multimodality models
> A1:This is an excellent question. We have observed a similar phenomenon of indistinguishable image features within the same category in our further experiments with the ALIGN model. We attribute this to potential induction biases that arise during the training process of large-scale multimodal models based on contrastive learning. For example, many images of dogs may have different textual descriptions associated with them during training, and the textual variations can cause the model to push their image features further apart, resulting in the phenomenon of inseparable image features in the classification task. Our method aims to mitigate this issue to some extent and achieve higher accuracy. We will further analyze and compare our approach with other multimodality models as suggested.

---

### Meta-Review · Area_Chair_akUs · 2023-12-08

**Metareview:**

This paper addresses domain generalization with CLIP models, proposing image encoder tuning (Im-Tuning) to enhance feature separability. While the approach is well-presented and effective, some concerns about the lack of in-depth analysis on the separability issue and limited comparisons to other multimodal models impact the overall contribution. Additionally, clarity in argumentation and potential efficiency concerns with increased parameters need attention. Further exploration of novel aspects and addressing weaknesses will strengthen the paper's significance in the context of domain generalization with CLIP models.

After rebuttal, none of the "negative" reviewers were convinced. The authors acknowledged that they failed to compare with more sota visual prompt tuning models and AC believes that it is not proper to ask the definitive question that "Prompt tuning is not all we need" before a comprehensive literature review.

**Justification For Why Not Higher Score:**

the paper raises a big question but the approach is easy and not yet outperforming other SOTAs.

**Justification For Why Not Lower Score:**

N/A.

---

### Decision · Program_Chairs · 2024-01-16

Reject